# Frailty does not cause all frail symptoms: United States Health and Retirement Study

**Yi-Sheng Chao** [1]*, **Chao-Jung Wu** [2], **June Y. T. Po** [3], **Shih-Yu Huang** [4,5], **Hsing-Chien Wu** [6], **Hui-Ting Hsu** [7], **Yen-Po Cheng** [7], **Yi-Chun Lai** [8], **Wei-Chih Chen** [9]*

**1** Independent Researcher, Montreal, Canada, **2** Université du Québec à Montréal, Montreal, Canada, **3** Natural Resources Institute, University of Greenwich, London, United Kingdom, **4** Department of Anesthesiology, Shuang Ho Hospital, Taipei Medical University, New Taipei City, Taiwan, **5** Department of Anesthesiology, School of Medicine, College of Medicine, Taipei Medical University, Taipei, Taiwan, **6** Taipei Hospital, Ministry of Health and Welfare, New Taipei City, Taiwan, **7** Changhua Christian Hospital, Changhua, Taiwan, **8** National Yang-Ming University Hospital, Yilan, Taiwan, **9** Department of Chest Medicine, Taipei Veterans General Hospital, Institute of Emergency and Critical Care Medicine, National Yang-Ming University, Taipei City, Taiwan

* chaoyisheng@post.harvard.edu (YSC); wiji.chen@gmail.com (WCC)

## Abstract

### Background

Frailty is associated with major health outcomes. However, the relationships between frailty and frailty symptoms haven't been well studied. This study aims to show the associations between frailty and frailty symptoms.

### Methods

The Health and Retirement Study (HRS) is an ongoing longitudinal biannual survey in the United States. Three of the most used frailty diagnoses, defined by the Functional Domains Model, the Burden Model, and the Biologic Syndrome Model, were reproduced according to previous studies. The associations between frailty statuses and input symptoms were assessed using odds ratios and correlation coefficients.

### Results

The sample sizes, mean ages, and frailty prevalence matched those reported in previous studies. Frailty statuses were weakly correlated with each other (coefficients = 0.19 to 0.38, p < 0.001 for all). There were 49 input symptoms identified by these three models. Frailty statuses defined by the three models were not significantly correlated with one or two symptoms defined by the same models (p > 0.05 for all). One to six symptoms defined by the other two models were not significantly correlated with each of the three frailty statuses (p > 0.05 for all). Frailty statuses were significantly correlated with their own bias variables (p < 0.05 for all).

### Conclusion

Frailty diagnoses lack significant correlations with some of their own frailty symptoms and some of the frailty symptoms defined by the other two models. This finding raises questions

**Data Availability Statement:** The HRS data produced by the RAND Center for the Study of Aging can be accessed via the University of Michigan site (https://hrs.isr.umich.edu/data-

products). The authors do not have special access to the HRS data.

**Funding:** This study was supported by Taipei Veterans General Hospital in the form of a grant awarded to WCC (V111B-024). The funder had no role in study design, data collection and analysis, decision to publish, or preparation of the manuscript. No additional external funding was received for this study.

**Competing interests:** YSC is employed by the Canadian Agency for Drugs and Technologies in Health. YSC conducted this study as an independent researcher out of academic curiosity without any material support. The funding agency had no role in this study. This study is not associated with any patents, products in development or marketed products. This does not alter our adherence to PLOS ONE policies on sharing data and materials.

like whether the frailty symptoms lacking significant correlations with frailty statuses could be included to diagnose frailty and whether frailty exists and causes frailty symptoms.

## Introduction

Frailty is a syndrome and can be diagnosed with composite criteria that consist of various frailty symptoms [1–3]. Frailty is often characterized by age-related symptoms, such as declines in physical and cognitive functioning. It has been considered significant for the prediction of major health outcomes, such as falls, surgical outcomes, and mortality [2, 3]. By aggregating information from multiple symptoms, frailty index scores can be assigned to individuals [2, 3]. Frailty status can then be derived by applying theoretical thresholds to frailty index scores [2, 3]. Three of the most commonly used frailty indices require 4 to 70 input domains or symptoms for the diagnosis of frailty [2].

Ideally, pathological changes or underlying health conditions are expected to cause or lead to significant increases in symptom occurrence. The increased frequency of symptom development can be used to make diagnoses that serve as proxy measures to the pathological changes {please add new reference: DOI:10.1038/s41598-022-14826-2}. For example, frailty has been recognized as a cause of disability, independent of clinical conditions [4]. Other researchers also found that frailty can lead to symptoms, particularly mental symptoms [5, 6] and fatigue [7]. Frailty has been confirmed using multiple frailty symptoms and considered a diagnosis that represent low physical reserve [8]. However, the effects of frailty on the development of frailty symptoms (those used to diagnose frailty) have not been well discussed. Instead, frailty has been described and defined differently [1]. Some studies have shown that how frailty is diagnosed seems far from ideal and lacks pathological confirmation [2]. Additionally, researchers have confirmed notable differences in the frail patients identified between the three frailty models [1].

In other words, whether frailty causes frailty symptoms remains unclear and whether frailty should cause frailty symptoms has not been explicitly declared in various, often conflicting, theories of frailty [2]. Causal relationships can be established through different approaches [9], such as using the Bradford-Hill Criteria [10]. Among all requirements in the Bradford-Hill Criteria, the strengths of associations between frailty and frailty symptoms are important and can be used to assess the impact of frailty prevention programs on frailty treatment and to understand the mechanisms that cause frailty. For example, it has been suggested that cognitive impairment plays an important role for frailty diagnosis and mortality among frail patients [3]. The confirmation of the causal relationship between frailty and cognitive function has the potential for intervention development. Without extensive reviews of the relationships between frailty and its symptoms, how frailty may influence frailty symptoms is an important question that is left unanswered. This study aims to assess the effect of frailty on the occurrence of frailty symptoms using a cohort that have been used to compare three of the most used frailty indices.

## Methods

The Health and Retirement Study (HRS) follows Americans aged 50 years and over every 2 years [1, 2, 11, 12]. The 2004 wave HRS data were used to compare frailty indices defined by 3 models: the Functional Domains Model by Strawbridge et al. [13], the Burden Model by Rockwood et al. [14, 15], and the Biologic Syndrome Model by Fried et al. [4]. Frailty symptoms or

input variables were used to defined various domains defined by the 3 models [2, 3, 16]. When individuals presented enough numbers of frailty symptoms in a domain, these individuals might be considered to have a deficit in this domain for the Functional Domain Model and the Biologic Syndrome Model [2]. For example, the weight loss domain in the Biologic Syndrome Model asked individuals whether they had body mass index (BMI) less than 18.5 kg/m$^2$ or whether they lost weight for 10% or more, compared to two years ago [1]. This domain required information on weights, heights, BMI, and weights two years ago [1].

In the Burden Model, one symptom represented a single domain and the presence of a symptom suggested the occurrence of a deficit [1, 2]. The frailty indices were the numbers of deficits identified using 4, 70, and 5 domains defined by the 3 frailty models, respectively [1]. Frailty statuses could be diagnosed when individuals had 2, 18 (70 times 0.25), or 3 deficits according to the 3 models, respectively [2]. The details of the frailty symptoms and input variables were published elsewhere (https://doi.org/10.1371/journal.pone.0197859.s002) [2]. The names and definitions of the input variables, frailty symptoms, and domains are listed in Tables 1–3. There were 10, 26, and 14 variables (frailty symptoms, input variables, or domains) identified for the 3 models, respectively. In total, there were 57 variables required to produce the frailty indices defined by the 3 models. In addition, there were 4 bias variables induced by the 4 domains in the Functional Domains Model (Table 1), 1 bias variable induced by the Burden Model (Table 2), and 4 bias variables induced by the Biologic Syndrome Model (Table 3) [2].

## Statistical analyses

The associations between the frailty statuses and their frailty symptoms were determined with odds ratios and correlation coefficients. Odds ratios were the ratios of the odds of developing symptoms occurred among frail individuals, compared to the odds among those not frail [17]. Odds ratios were applicable to binomial variables [17]. Odds ratios equaling 1 suggest that the two groups have similar risks of developing symptoms [17]. The processing to transform non-binomial variables to binomial variables were according to the recommendations by the authors of the Burden Model [18]. Pearson's correlation coefficients were used to assess the associations between frailty statuses defined by the 3 models and frailty symptoms or input variables or domains or bias variables [19]. Correlation coefficients ranged from -1 to 1, representing completely opposite information and identical information between 2 variables, respectively. We hypothesized that 1) frailty statuses were not associated with symptom incidence (odds ratio = 1); 2) frailty statuses were not correlated with frailty symptoms or input variables of the frailty indices (correlation coefficient = 0). Correlation coefficients between 0 and 0.10, 0.10 and 0.39, 0.40 and 0.69, 0.70 and 0.89, and 0.90 and 1.00 were interpreted as negligible, weak, moderate, strong, and very strong correlations, respectively [20]. P values were adjusted for multiple comparison using false discovery rates [21]. Two-tailed P values that were less than 0.05 were considered statistically significant. All statistical analyses were conducted within R environment (v4.0.4) [22] and RStudio (v1.4.1106) [23]. This secondary data analysis was approved by the ethics review committee at the Centre Hospitalier de l'Université de Montréal.

## Results

There were 11,113, 7,713, and 1,642 HRS participants analyzed for the frailty indices defined by the Functional Domains Model, the Burden Model, and the Biologic Syndrome Model in Tables 1–3, respectively [2]. The numbers of frail patients were 3,059 (27.53%), 3,442 (44.63%), and 203 (12.36%), respectively [2]. The mean ages were 74.92, 78.43, and 77.05 years, respectively. The proportions of females were 57.46%, 58.78%, and 54.69%, respectively.

**Table 1. Frailty status defined by the Functional Domains Model and its associations with frailty symptoms.**

| HRS variables | Definitions | N with symptoms for binomial variables | Odds ratios (95% CIs | Mean (SD) | Correlation coefficients (95% CIs) |
|---|---|---|---|---|---|
| **r7agey_b** | Age at interview (years) | | | 73.92 (7.59) | 0.26 (0.24 to 0.27)*** |
| **r7bmi** | Self-reported body mass index = kg/m2 | | | 26.68 (5.31) | -0.02 (-0.04 to 0) |
| **r7cogtot** | Total cognition summary score | | | 20.53 (6.08) | -0.41 (-0.42 to -0.39)*** |
| **r7dizz** | Physical functioning: Dizziness as persistent problem | 1577 | 7.9 (7.03 to 8.87)*** | 0.14 (0.35) | 0.36 (0.35 to 0.38)*** |
| **r7eye** | Sensory problems: Fair or poor eyesight despite use of corrective lenses | | | 2.95 (1.01) | 0.44 (0.42 to 0.45)*** |
| **r7fall** | fallen down last 2 years | | | 0.9 (2.88) | 0.29 (0.27 to 0.31)*** |
| **r7frailim1** | Frailty index: Functional Domains Model | | | 0.98 (0.93) | 0.85 (0.84 to 0.85)*** |
| **r7frailim1cat** | Frailty status: Functional Domains Model (outcome of this table) | 3059 | Not applicable | 0.28 (0.45) | 1 (1 to 1)*** |
| **r7hear** | Sensory problems: fair or poor hearing despite use of hearing aides | | | 2.87 (1.12) | 0.37 (0.35 to 0.39)*** |
| **r7lift** | some difficulty in lift/carry 10lbs | 3631 | 8.85 (8.05 to 9.72)*** | 0.33 (0.47) | 0.46 (0.45 to 0.48)*** |
| **r7memopr** | Proxy memory rating | | | 1.39 (1.18) | 0.33 (0.31 to 0.35)*** |
| **r7wchange** | Weight in wave 2002 minus weight in wave 2004 (%) | | | 0.01 (0.08) | -0.04 (-0.05 to -0.02)*** |
| **ragender** | Male = 0; female = 1 | 6385 | 1.32 (1.21 to 1.44)*** | 0.57 (0.49) | 0.06 (0.04 to 0.08)*** |
| **Domains and other frailty symptoms identified by the other 2 models** | | | | | |
| **r7actsum** | Summary scores of physical activities | | | 33.45 (9.11) | 0.31 (0.29 to 0.33)*** |
| **r7arthrcat** | Binomial: Arthritis | 7693 | 1.84 (1.67 to 2.03)*** | 0.69 (0.46) | 0.12 (0.1 to 0.14)*** |
| **r7bathcat** | Binomial: Problems with bathing | 1264 | 8.28 (7.27 to 9.42)*** | 0.11 (0.32) | 0.34 (0.32 to 0.36)*** |
| **r7cancrcat** | Binomial: Malignant disease | 1978 | 1.03 (0.92 to 1.15) | 0.18 (0.38) | 0.01 (-0.01 to 0.02) |
| **r7cogimpair** | Impaired cognition based on performance-based scores or proxy assessment | 999 | 23.09 (19.14 to 27.85)*** | 0.09 (0.29) | 0.41 (0.4 to 0.43)*** |
| **r7deprescat** | Binomial: Feeling sad, blue, depressed | 2011 | 3.54 (3.2 to 3.91)*** | 0.18 (0.39) | 0.24 (0.22 to 0.26)*** |
| **r7diabscat** | Binomial: History of diabetes mellitus | 292 | 1.38 (1.08 to 1.77)* | 0.03 (0.16) | 0.02 (0.01 to 0.04)** |
| **r7dresscat** | Binomial: problem getting dressed | 1419 | 6.01 (5.34 to 6.76)*** | 0.13 (0.33) | 0.31 (0.29 to 0.32)*** |
| **r7effort** | everything an effort | 2932 | 3.96 (3.62 to 4.34)*** | 0.26 (0.44) | 0.29 (0.27 to 0.31)*** |
| **r7fall_cat1** | More than 1 falls | 3640 | 3.6 (3.3 to 3.93)*** | 0.33 (0.47) | 0.28 (0.26 to 0.3)*** |
| **r7fall_cat2** | More than 2 falls | 1929 | 6.71 (6.04 to 7.46)*** | 0.17 (0.38) | 0.36 (0.35 to 0.38)*** |
| **r7frail1_1** | Dizziness as persistent problem, > = 2 falls in previous 2 years, or difficulty lifting 10 pounds | 4383 | 27.81 (24.63 to 31.42)*** | 0.39 (0.49) | 0.61 (0.6 to 0.62)*** |
| **r7frail1_2** | Weight in wave 2002 minus weight in wave 2004! 10% of weight in wave 2002 or body mass index o18.5 kg/m2 | 866 | 9.35 (7.97 to 10.96)*** | 0.08 (0.27) | 0.3 (0.29 to 0.32)*** |

*(Continued)*

**Table 1.** (Continued)

| HRS variables | Definitions | N with symptoms for binomial variables | Odds ratios (95% CIs | Mean (SD) | Correlation coefficients (95% CIs) |
|---|---|---|---|---|---|
| **r7frail1_3** | Mild to severe cognitive impairment on performance-based measure or according to proxy and interviewer rating | | | 0.09 (0.29) | 0.42 (0.4 to 0.43)*** |
| **r7frail1_4** | Fair or poor eyesight despite use of corrective lenses or fair or poor hearing despite use of hearing aides | | | 0.42 (0.49) | 0.59 (0.58 to 0.6)*** |
| **r7frail3_2** | Yes to either of two CES-D items: (i) Felt that everything I did was an effort in last week. (ii) Could not get going in last week. | 4025 | 3.99 (3.66 to 4.35)*** | 0.36 (0.48) | 0.3 (0.29 to 0.32)*** |
| **r7frail3_3** | Frequency of three intensities of activity, lowest quintile (stratified according to sex) | 2872 | 4.11 (3.75 to 4.51)*** | 0.26 (0.44) | 0.3 (0.28 to 0.32)*** |
| **r7frail3_4** | Time to walk 8 ft, converted to time to walk 15 ft. Cutoff criteria according to sex and height remain the same | 5474 | 1.44 (1.33 to 1.57)*** | 0.49 (0.5) | 0.08 (0.06 to 0.1)*** |
| **r7frail3_5** | Grip strength: Weakest 20% (stratified according to sex and BMI) | 2529 | 1.43 (1.3 to 1.57)*** | 0.23 (0.42) | 0.07 (0.05 to 0.09)*** |
| **r7frailim1** | Frailty index: Functional Domains Model | | | 0.98 (0.93) | 0.85 (0.84 to 0.85)*** |
| **r7frailim2** | Frailty index: Burden Model | | | 5.02 (2.83) | 0.46 (0.44 to 0.47)*** |
| **r7frailim2cat** | Frailty status: Burden Model | | | 0.45 (0.5) | 0.38 (0.36 to 0.4)*** |
| **r7frailim3** | Frailty index: Biologic Syndrome Model | | | 1.14 (1.08) | 0.35 (0.3 to 0.39)*** |
| **r7frailim3cat** | Frailty status: Biologic Syndrome Model | | | 0.12 (0.33) | 0.3 (0.26 to 0.34)*** |
| **r7going** | Could not get going | 2682 | 3.41 (3.11 to 3.73)*** | 0.24 (0.43) | 0.25 (0.24 to 0.27)*** |
| **r7grip** | Grip strength, largest value | | | 30.19 (16.32) | -0.07 (-0.09 to -0.06)*** |
| **r7gripl** | Grip strength, left hand | | | 26.63 (16.29) | -0.07 (-0.09 to -0.05)*** |
| **r7gripr** | Grip strength, right hand | | | 28.74 (16.14) | -0.1 (-0.12 to -0.08)*** |
| **r7headac** | Headache | 817 | 3.05 (2.64 to 3.53)*** | 0.07 (0.26) | 0.15 (0.13 to 0.17)*** |
| **r7heartcat** | Binomial: Cardiac problems | 3631 | 2.15 (1.97 to 2.35)*** | 0.33 (0.47) | 0.17 (0.15 to 0.18)*** |
| **r7height** | Self-reported height in meters | | | 1.68 (0.1) | -0.09 (-0.11 to -0.07)*** |
| **r7hibp** | had high blood pressure since last interview | 6791 | 1.49 (1.36 to 1.62)*** | 0.61 (0.49) | 0.08 (0.07 to 0.1)*** |
| **r7ltactx** | Frequency of light physical activity | | | 2.87 (1.2) | 0.29 (0.28 to 0.31)*** |
| **r7lungcat** | Binomial: Lung problems | 1345 | 2.02 (1.79 to 2.27)*** | 0.12 (0.33) | 0.11 (0.09 to 0.13)*** |
| **r7mdactx** | Frequency of moderate physical activity | | | 3.13 (1.37) | 0.29 (0.27 to 0.31)*** |
| **r7memryscat** | Binomial: Memory changes | 345 | 7.46 (5.86 to 9.48)*** | 0.03 (0.17) | 0.18 (0.16 to 0.2)*** |
| **r7mobila** | Some difficulty in mobility /05 | | | 1.36 (1.59) | 0.43 (0.42 to 0.45)*** |
| **r7muscle** | Musculoskeletal problems | 386 | 1.3 (1.05 to 1.62)* | 0.03 (0.18) | 0.02 (0 to 0.04)* |
| **r7psychcat** | Binomial: Depression | 1799 | 2.82 (2.55 to 3.13)*** | 0.16 (0.37) | 0.19 (0.17 to 0.21)*** |
| **r7psychscat** | Binomial: Changes in general mental functioning | 246 | 3.6 (2.78 to 4.64)*** | 0.02 (0.15) | 0.1 (0.08 to 0.12)*** |

*(Continued)*

**Table 1.** (Continued)

| HRS variables | Definitions | N with symptoms for binomial variables | Odds ratios (95% CIs | Mean (SD) | Correlation coefficients (95% CIs) |
|---|---|---|---|---|---|
| **r7seizure** | Seizures, generalized | Not available | | | |
| **r7sleeprcat** | Binomial: Sleep changes | 3102 | 2.26 (2.07 to 2.47)*** | 0.28 (0.45) | 0.17 (0.15 to 0.19)*** |
| **r7strokcat** | Binomial: Cerebrovascular problems | 1139 | 3.36 (2.97 to 3.81)*** | 0.1 (0.3) | 0.19 (0.17 to 0.21)*** |
| **r7strokecat** | Binomial: History of stroke | 1283 | 3.39 (3.01 to 3.82)*** | 0.12 (0.32) | 0.2 (0.18 to 0.22)*** |
| **r7stroks** | Had stroke since last interview | 255 | 3.31 (2.58 to 4.25)*** | 0.02 (0.15) | 0.09 (0.08 to 0.11)*** |
| **r7tired** | Tiredness all the time | 1 | Not applicable | 0 (0.01) | -0.01 (-0.02 to 0.01) |
| **r7toiltcat** | Binomial: Toileting problems | 903 | 6.9 (5.95 to 8)*** | 0.08 (0.27) | 0.27 (0.26 to 0.29)*** |
| **r7underw** | Underweight in wave 2004 (%) | 313 | 7.94 (6.15 to 10.25)*** | 0.03 (0.17) | 0.18 (0.16 to 0.19)*** |
| **r7urine** | Urinary incontinence | 2710 | 2.35 (2.15 to 2.58)*** | 0.24 (0.43) | 0.18 (0.16 to 0.19)*** |
| **r7vgactx** | Frequency of vigorous physical activity | | | 4.24 (1.24) | 0.2 (0.18 to 0.22)*** |
| **r7walkt** | Slowness: Time to walk 8 ft, converted to time to walk 15 ft. Cutoff criteria according to sex and height remain the same | | | 5.09 (19.08) | 0.11 (0.1 to 0.13)*** |
| **r7walkt15** | Time to walk 15 feet | | | 9.54 (35.77) | 0.11 (0.1 to 0.13)*** |
| **r7weight** | Self-reported weight in kilograms | | | 76.03 (17.43) | -0.06 (-0.08 to -0.04)*** |
| **Bias variables** | | | | | |
| **Biases induced by the Functional Domains Model** | | | | | |
| **r7frail1_1res** | Bias induced by Dizziness as persistent problem, > = 2 falls in previous 2 years, or difficulty lifting 10 pounds | | | 0 (0.29) | 0.35 (0.34 to 0.37)*** |
| **r7frail1_2res** | Bias induced by Weight in wave 2002 minus weight in wave 2004! 10% of weight in wave 2002 or body mass index o18.5 kg/m2 | | | 0 (0.25) | 0.2 (0.19 to 0.22)*** |
| **r7frail1_3res** | Bias induced by Mild to severe cognitive impairment on performance-based measure or according to proxy and interviewer rating | | | 0 (0.15) | -0.03 (-0.05 to -0.01)** |
| **r7frail1_4res** | Bias induced by Fair or poor eyesight despite use of corrective lenses or fair or poor hearing despite use of hearing aides | | | 0 (0.2) | 0.02 (0.01 to 0.04)* |
| **Biases induced by the Burden Model** | | | | | |
| **r7frail2res** | Bias induced by Summary of proxy memory rating and total cognition summary score | | | 0 (0.04) | -0.14 (-0.16 to -0.11)*** |
| **Biases induced by the Biologic Syndrome Model** | | | | | |
| **r7frail3_2res** | Bias induced by Yes to either of two CES-D items: (i) Felt that everything I did was an effort in last week. (ii) Could not get going in last week. | | | 0 (0.17) | 0.01 (-0.04 to 0.06) |
| **r7frail3_3res** | Bias induced by Frequency of three intensities of activity, lowest quintile (stratified according to sex) | | | 0 (0.31) | 0.1 (0.05 to 0.15)*** |
| **r7frail3_4res** | Bias induced by Time to walk 8 ft, converted to time to walk 15 ft. Cutoff criteria according to sex and height remain the same | | | 0 (0.49) | 0.16 (0.12 to 0.21)*** |
| **r7frail3_5res** | Bias induced by Grip strength: Weakest 20% (stratified according to sex and BMI) | | | 0 (0.34) | 0.05 (0 to 0.09) |

n = 11,113; frailty n (%) = 3,059 (27.53%); mean age = 74.92 years; female % = 57.46%.

BMI = body mass index; CES-D = Center for Epidemiological Studies Depression; HRS = Health and Retirement Study.

* = p < 0.05

** = p < 0.01

*** = p < 0.001.

### Frailty symptom development based on frailty status

In Tables 1–3, the associations between frailty status (yes or no) and symptom development are shown using odds ratios and correlation coefficients. Overall, most of frailty symptoms were significantly associated with frailty statuses. However, frailty statuses defined by the three models were not significantly associated with all frailty symptoms or input variables or domains. The frailty symptoms or input variables or domains that were not significantly associated with frailty statuses are described below. The correlation coefficients between the three frailty statuses ranged from 0.19 to 0.38 (weak correlations, $p < 0.001$ for all).

In Table 1, the frailty status defined by the Functional Domains Model was not significantly correlated with one input variable, BMI (correlation coefficients = -0.02, 95% CI = -0.04 to 0).

Among the frailty symptoms or input variables or domains identified by the other two models, two symptoms, malignant disease and tiredness all the time, were not significantly associated with this frailty status (p of correlations $> 0.05$ for both symptoms). Among the bias variables, one bias variable that was induced by having one of two Center for Epidemiologic Studies Depression (CES-D) items was not significantly associated with this frailty status (correlation $p > 0.05$).

In Table 2, the frailty status defined by the Burden Model was assessed for the associations with frailty symptoms, input variables, and domains. One input symptom, tiredness all the time, was not significantly associated with this frailty status (p of correlation $> 0.05$). One symptom identified by the other two models, self-reported weight, was not significantly correlated with this frailty status ($p > 0.05$). Among the bias variables, four were not significantly correlated with this frailty status ($p > 0.05$ for all).

In Table 3, the frailty status defined by the Biologic Syndrome Model was assessed for the associations with frailty symptoms, input variables, and domains. BMI was not significantly correlated with this frailty status ($p > 0.05$). Because the values of two symptoms, proxy memory rating and history of stroke, were the same for frail and non-frail HRS participants for this frailty index, their correlations with this frailty status could not be assessed. Six symptoms defined by the other two models, history of malignant disease, diabetes mellitus, headache, memory change, musculoskeletal problems, and change in general mental functioning, were not significantly correlated with this frailty status ($p > 0.05$ for all).

### Correlations with bias variables

In Tables 1–3, the correlations with bias variables are shown for the three frailty indices. Each frailty status was significantly associated with the bias variables induced by their own diagnostic criteria. The frailty statuses defined by the Functional Domains Model, the Burden Model, and the Biologic Syndrome Model, were significantly correlated with four, one, and four bias variables induced by their own models, respectively ($p < 0.05$ for all). In addition, the frailty status defined by the Functional Domains Model, the Burden Model, and the Biologic Syndrome Model, were significantly correlated with three, four, and two bias variables induced by the other two models, respectively ($p < 0.05$ for all).

### Discussion

Strengths of the associations are one of the Bradford-Hill criteria to assess whether a disease causes symptoms or outcomes [10]. Frailty has been promising in establishing causal relationships with major health outcomes, such as mortality and falls, based on frailty's significant associations with them [2]. However, whether frailty should cause frailty symptoms has not been declared in the theories of frailty and whether frailty causes frailty symptoms have not been well studied. In this study using the HRS data, three of the most used frailty diagnoses fail

**Table 2. Frailty status defined by the Burden Model and its associations with frailty symptoms.**

| HRS variables | Definitions | N with symptoms for binomial variables | Odds ratios (95% CIs | Mean (SD) | Correlation coefficients (95% CIs) |
|---|---|---|---|---|---|
| r7agey_b | Age at interview (years) | | | 77.43 (6.46) | 0.19 (0.17 to 0.22)*** |
| r7arthrcat | Binomial: Arthritis | 5457 | 4.45 (3.97 to 4.99)*** | 0.71 (0.45) | 0.3 (0.28 to 0.32)*** |
| r7bathcat | Binomial: Problems with bathing | 1091 | 28.44 (21.95 to 36.86)*** | 0.14 (0.35) | 0.4 (0.39 to 0.42)*** |
| r7cancrcat | Binomial: Malignant disease | 1489 | 2.12 (1.89 to 2.38)*** | 0.19 (0.39) | 0.15 (0.13 to 0.17)*** |
| r7cogtot | Total cognition summary score | | | 19.69 (6.27) | -0.25 (-0.27 to -0.23)*** |
| r7deprescat | Binomial: Feeling sad, blue, depressed | 1480 | 5.55 (4.87 to 6.32)*** | 0.19 (0.39) | 0.31 (0.29 to 0.33)*** |
| r7diabscat | Binomial: History of diabetes mellitus | 174 | 3.07 (2.21 to 4.26)*** | 0.02 (0.15) | 0.08 (0.06 to 0.1)*** |
| r7dresscat | Binomial: problem getting dressed | 1159 | 18.97 (15.34 to 23.48)*** | 0.15 (0.36) | 0.4 (0.38 to 0.42)*** |
| r7fall | Fallen down last 2 years | | | 0.99 (2.89) | 0.3 (0.28 to 0.32)*** |
| r7frailim2 | Frailty index: Burden Model | | | 5.02 (2.83) | 0.8 (0.8 to 0.81)*** |
| r7frailim2cat | Frailty status: Burden Model (outcome of this table) | 3442 | Not applicable | 0.45 (0.5) | 1 (1 to 1)*** |
| r7headac | Headache | 514 | 5.28 (4.25 to 6.58)*** | 0.07 (0.25) | 0.19 (0.17 to 0.21)*** |
| r7heartcat | Binomial: Cardiac problems | 2817 | 4.01 (3.64 to 4.42)*** | 0.37 (0.48) | 0.32 (0.3 to 0.34)*** |
| r7hibp | Had high blood pressure since last interview | 4809 | 2.81 (2.55 to 3.1)*** | 0.62 (0.48) | 0.24 (0.22 to 0.26)*** |
| r7lungcat | Binomial: Lung problems | 928 | 4.41 (3.77 to 5.16)*** | 0.12 (0.33) | 0.23 (0.2 to 0.25)*** |
| r7memopr | Proxy memory rating | | | 1.45 (1.29) | 0.23 (0.21 to 0.25)*** |
| r7memryscat | Binomial: Memory changes | 303 | 5.95 (4.43 to 8)*** | 0.04 (0.19) | 0.15 (0.13 to 0.17)*** |
| r7mobila | Some difficulty in mobility /05 | | | 1.51 (1.66) | 0.56 (0.55 to 0.58)*** |
| r7muscle | Musculoskeletal problems | 258 | 3.61 (2.73 to 4.78)*** | 0.03 (0.18) | 0.11 (0.09 to 0.13)*** |
| r7psychcat | Binomial: Depression | 1230 | 7.74 (6.63 to 9.04)*** | 0.16 (0.37) | 0.33 (0.31 to 0.35)*** |
| r7psychscat | Binomial: Changes in general mental functioning | 188 | 15.02 (8.84 to 25.5)*** | 0.02 (0.15) | 0.15 (0.13 to 0.17)*** |
| r7seizure | Seizures, generalized | 0 | Not applicable | 0 | Not applicable for uniform values |
| r7sleeprcat | Binomial: Sleep changes | 2185 | 5.05 (4.53 to 5.63)*** | 0.28 (0.45) | 0.35 (0.33 to 0.37)*** |
| r7strokcat | Binomial: Cerebrovascular problems | 923 | 11.45 (9.34 to 14.03)*** | 0.12 (0.32) | 0.32 (0.3 to 0.34)*** |
| r7strokecat | Binomial: History of stroke | 1062 | 10.42 (8.68 to 12.51)*** | 0.14 (0.34) | 0.34 (0.32 to 0.36)*** |
| r7tired | Tiredness all the time | 1 | Not applicable | 0 (0.01) | 0.01 (-0.01 to 0.03) |
| r7toiltcat | Binomial: Toileting problems | 759 | 44.18 (29.79 to 65.52)*** | 0.1 (0.3) | 0.35 (0.33 to 0.36)*** |

*(Continued)*

**Table 2.** (Continued)

| HRS variables | Definitions | N with symptoms for binomial variables | Odds ratios (95% CIs) | Mean (SD) | Correlation coefficients (95% CIs) |
|---|---|---|---|---|---|
| r7urine | Urinary incontinence | 2075 | 6.41 (5.71 to 7.19)*** | 0.27 (0.44) | 0.38 (0.36 to 0.4)*** |
| ragender | Male = 0; female = 1 | 4534 | 1.69 (1.54 to 1.85)*** | 0.59 (0.49) | 0.13 (0.1 to 0.15)*** |
| **Domains and other frailty symptoms identified by the other 2 models** | | | | | |
| r7actsum | Summary scores of physical activities | | | 34.44 (8.92) | 0.35 (0.33 to 0.37)*** |
| r7bmi | Self-reported body mass index = kg/m2 | | | 26.1 (5.07) | 0.07 (0.05 to 0.1)*** |
| r7cogimpair | Impaired cognition based on performance-based scores or proxy assessment | 884 | 4.71 (4 to 5.54)*** | 0.11 (0.32) | 0.23 (0.21 to 0.25)*** |
| r7dizz | Physical functioning: Dizziness as persistent problem | 1163 | 3.93 (3.42 to 4.5)*** | 0.15 (0.36) | 0.23 (0.21 to 0.25)*** |
| r7effort | Everything an effort | 2153 | 4.59 (4.12 to 5.11)*** | 0.28 (0.45) | 0.33 (0.31 to 0.35)*** |
| r7eye | Sensory problems: Fair or poor eyesight despite use of corrective lenses | | | 3 (1.03) | 0.26 (0.24 to 0.28)*** |
| r7fall_cat1 | More than 1 falls | 2761 | 5.61 (5.06 to 6.21)*** | 0.36 (0.48) | 0.39 (0.37 to 0.41)*** |
| r7fall_cat2 | More than 2 falls | 1499 | 6.94 (6.06 to 7.95)*** | 0.19 (0.4) | 0.35 (0.33 to 0.37)*** |
| r7frail1_1 | Dizziness as persistent problem, > = 2 falls in previous 2 years, or difficulty lifting 10 pounds | 3290 | 6.02 (5.45 to 6.64)*** | 0.43 (0.49) | 0.42 (0.4 to 0.44)*** |
| r7frail1_2 | Weight in wave 2002 minus weight in wave 2004! 10% of weight in wave 2002 or body mass index o18.5 kg/m2 | 635 | 1.97 (1.67 to 2.32)*** | 0.08 (0.27) | 0.09 (0.07 to 0.11)*** |
| r7frail1_3 | Mild to severe cognitive impairment on performance-based measure or according to proxy and interviewer rating | | | 0.11 (0.32) | 0.23 (0.21 to 0.25)*** |
| r7frail1_4 | Fair or poor eyesight despite use of corrective lenses or fair or poor hearing despite use of hearing aides | | | 0.45 (0.5) | 0.22 (0.2 to 0.24)*** |
| r7frail3_2 | Yes to either of two CES-D items: (i) Felt that everything I did was an effort in last week. (ii) Could not get going in last week. | 2977 | 4.72 (4.28 to 5.21)*** | 0.39 (0.49) | 0.36 (0.34 to 0.38)*** |
| r7frail3_3 | Frequency of three intensities of activity, lowest quintile (stratified according to sex) | 2226 | 4.89 (4.39 to 5.45)*** | 0.29 (0.45) | 0.34 (0.32 to 0.36)*** |
| r7frail3_4 | Time to walk 8 ft, converted to time to walk 15 ft. Cutoff criteria according to sex and height remain the same | 3853 | 1.37 (1.25 to 1.5)*** | 0.5 (0.5) | 0.08 (0.06 to 0.1)*** |
| r7frail3_5 | Grip strength: Weakest 20% (stratified according to sex and BMI) | 1840 | 1.4 (1.26 to 1.55)*** | 0.24 (0.43) | 0.07 (0.05 to 0.09)*** |
| r7frailim1 | Frailty index: Functional Domains Model | | | 1.08 (0.96) | 0.43 (0.41 to 0.45)*** |
| r7frailim1cat | Frailty status: Functional Domains Model | 2434 | 5.57 (5.01 to 6.19)*** | 0.32 (0.46) | 0.38 (0.36 to 0.4)*** |
| r7frailim2 | Frailty index: Burden Model | | | 5.02 (2.83) | 0.8 (0.8 to 0.81)*** |
| r7frailim3 | Frailty index: Biologic Syndrome Model | | | 1.22 (1.11) | 0.28 (0.23 to 0.33)*** |
| r7frailim3cat | Frailty status: Biologic Syndrome Model | | | 0.14 (0.35) | 0.19 (0.14 to 0.24)*** |
| r7going | could not get going | 2026 | 4.64 (4.15 to 5.18)*** | 0.26 (0.44) | 0.32 (0.3 to 0.34)*** |
| r7grip | Grip strength, largest value | | | 29.49 (15.93) | -0.08 (-0.1 to -0.06)*** |
| r7gripl | Grip strength, left hand | | | 25.89 (15.72) | -0.07 (-0.09 to -0.04)*** |
| r7gripr | Grip strength, right hand | | | 27.99 (15.37) | -0.11 (-0.13 to -0.09)*** |

(Continued)

**Table 2.** (Continued)

| HRS variables | Definitions | N with symptoms for binomial variables | Odds ratios (95% CIs | Mean (SD) | Correlation coefficients (95% CIs) |
|---|---|---|---|---|---|
| r7hear | Sensory problems: fair or poor hearing despite use of hearing aides | | | 2.97 (1.13) | 0.16 (0.14 to 0.18)*** |
| r7height | Self-reported height in meters | | | 1.68 (0.1) | -0.1 (-0.13 to -0.08)*** |
| r7lift | Some difficulty in lift/carry 10lbs | 2862 | 5.88 (5.31 to 6.51)*** | 0.37 (0.48) | 0.4 (0.39 to 0.42)*** |
| r7ltactx | Frequency of light physical activity | | | 2.99 (1.24) | 0.3 (0.28 to 0.32)*** |
| r7mdactx | Frequency of moderate physical activity | | | 3.24 (1.41) | 0.34 (0.32 to 0.36)*** |
| r7stroks | Had stroke since last interview | 208 | 13 (8.09 to 20.88)*** | 0.03 (0.16) | 0.15 (0.13 to 0.18)*** |
| r7underw | Underweight in wave 2004 (%) | 270 | 1.68 (1.31 to 2.14)*** | 0.04 (0.18) | 0.05 (0.03 to 0.07)*** |
| r7vgactx | Frequency of vigorous physical activity | | | 4.35 (1.19) | 0.23 (0.21 to 0.25)*** |
| r7walkt | Slowness: Time to walk 8 ft, converted to time to walk 15 ft. Cutoff criteria according to sex and height remain the same | | | 5.2 (19.49) | 0.12 (0.1 to 0.15)*** |
| r7walkt15 | Time to walk 15 feet | | | 9.75 (36.55) | 0.12 (0.1 to 0.15)*** |
| r7wchange | Weight in wave 2002 minus weight in wave 2004 (%) | | | 0.01 (0.08) | 0.04 (0.02 to 0.07)*** |
| r7weight | Self-reported weight in kilograms | | | 73.91 (16.73) | 0 (-0.02 to 0.03) |
| **Bias variables** | | | | | |
| **Bias variables induced by the Functional Domains Model** | | | | | |
| r7frail1_1res | Bias induced by Dizziness as persistent problem, > = 2 falls in previous 2 years, or difficulty lifting 10 pounds | | | 0.01 (0.3) | 0.19 (0.17 to 0.21)*** |
| r7frail1_2res | Bias induced by Weight in wave 2002 minus weight in wave 2004! 10% of weight in wave 2002 or body mass index o18.5 kg/m2 | | | 0 (0.26) | 0.15 (0.13 to 0.17)*** |
| r7frail1_3res | Bias induced by Mild to severe cognitive impairment on performance-based measure or according to proxy and interviewer rating | | | 0 (0.17) | -0.01 (-0.03 to 0.01) |
| r7frail1_4res | Bias induced by Fair or poor eyesight despite use of corrective lenses or fair or poor hearing despite use of hearing aides | | | 0 (0.21) | 0.01 (-0.01 to 0.03) |
| **Bias variables induced by the Burden Model** | | | | | |
| r7frail2res | Bias induced by Summary of proxy memory rating and total cognition summary score | | | 0 (0.04) | -0.06 (-0.08 to -0.04)*** |
| **Bias variables induced by the Biologic Syndrome Model** | | | | | |
| r7frail3_2res | Bias induced by Yes to either of two CES-D items: (i) Felt that everything I did was an effort in last week. (ii) Could not get going in last week. | | | 0 (0.17) | -0.04 (-0.09 to 0.01) |
| r7frail3_3res | Bias induced by Frequency of three intensities of activity, lowest quintile (stratified according to sex) | | | 0 (0.32) | 0.12 (0.07 to 0.17)*** |
| r7frail3_4res | Bias induced by Time to walk 8 ft, converted to time to walk 15 ft. Cutoff criteria according to sex and height remain the same | | | 0.03 (0.49) | 0.09 (0.03 to 0.14)** |
| r7frail3_5res | Bias induced by Grip strength: Weakest 20% (stratified according to sex and BMI) | | | 0.01 (0.35) | 0.03 (-0.03 to 0.08) |

n = 7,713; frailty n (%) = 6,755 (87.58%); mean age = 78.43 years; female % = 58.78%.

BMI = body mass index; CES-D = Center for Epidemiological Studies Depression; HRS = Health and Retirement Study.

\* = p < 0.05

\*\* = p < 0.01

\*\*\* = p < 0.001.

**Table 3. Frailty status defined by the Biologic Syndrome Model and its associations with frailty symptoms.**

| HRS | Definitions variables | N with symptoms for binomial variables | Odds ratios (95% CIs | Mean (SD) | Correlation coefficients (95% CIs) |
|---|---|---|---|---|---|
| **r7agey_b** | Age at interview (years) | | | 76.05 (7.36) | 0.23 (0.19 to 0.28)*** |
| **r7bmi** | Self-reported body mass index = kg/m2 | | | 26.41 (4.91) | -0.04 (-0.09 to 0.01) |
| **r7cogtot** | Total cognition summary score | | | 21.44 (4.79) | -0.22 (-0.26 to -0.17)*** |
| **r7effort** | Everything an effort | 258 | 5.56 (4.04 to 7.66)*** | 0.16 (0.36) | 0.28 (0.23 to 0.32)*** |
| **r7frailim3** | Frailty index: Biologic Syndrome Model | | | 1.14 (1.08) | 0.73 (0.7 to 0.75)*** |
| **r7frailim3cat** | Frailty status: Biologic Syndrome Model (outcome of this table) | 203 | Not applicable | 0.12 (0.33) | 1 (1 to 1)*** |
| **r7going** | Could not get going | 306 | 6.43 (4.7 to 8.8)*** | 0.19 (0.39) | 0.31 (0.27 to 0.36)*** |
| **r7gripl** | Grip strength, left hand | | | 24.85 (13.1) | -0.28 (-0.32 to -0.23)*** |
| **r7gripr** | Grip strength, right hand | | | 27.38 (13.46) | -0.32 (-0.36 to -0.28)*** |
| **r7height** | Self-reported height in meters | | | 1.69 (0.1) | -0.12 (-0.17 to -0.08)*** |
| **r7mdactx** | Frequency of moderate physical activity | | | 2.93 (1.3) | 0.41 (0.37 to 0.45)*** |
| **r7memopr** | Proxy memory rating (1 to 6) | | | 1 (0) | Not applicable for uniform values |
| **r7stroks** | had stroke since last interview (1 = no, 2 = yes) | | | 1 (0) | Not applicable for uniform values |
| **r7vgactx** | Frequency of vigorous physical activity | | | 4.18 (1.26) | 0.21 (0.16 to 0.26)*** |
| **r7walkt** | Slowness: Time to walk 8 ft, converted to time to walk 15 ft. Cutoff criteria according to sex and height remain the same | | | 4.14 (13.29) | 0.42 (0.38 to 0.46)*** |
| **ragender** | Male = 0; female = 1 | 898 | 2.61 (1.88 to 3.64)*** | 0.55 (0.5) | 0.14 (0.1 to 0.19)*** |
| **Domains and frailty symptoms identified by the other 2 models** | | | | | |
| **r7actsum** | Summary scores of physical activities | | | 32.41 (8.61) | 0.37 (0.33 to 0.41)*** |
| **r7arthrcat** | Binomial: Arthritis | 1140 | 2.4 (1.63 to 3.52)*** | 0.69 (0.46) | 0.11 (0.06 to 0.16)*** |
| **r7bathcat** | Binomial: Problems with bathing | 88 | 6.75 (4.3 to 10.59)*** | 0.05 (0.23) | 0.23 (0.18 to 0.28)*** |
| **r7cancrcat** | Binomial: Malignant disease | 300 | 1.28 (0.89 to 1.83) | 0.18 (0.39) | 0.03 (-0.02 to 0.08) |
| **r7cogimpair** | Impaired cognition based on performance-based scores or proxy assessment | 45 | 5.09 (2.75 to 9.42)*** | 0.03 (0.16) | 0.14 (0.09 to 0.19)*** |
| **r7deprescat** | Binomial: Feeling sad, blue, depressed | 0 | Not applicable | 0 | Not applicable for uniform values |
| **r7diabscat** | Binomial: History of diabetes mellitus | 42 | 1.7 (0.77 to 3.72) | 0.03 (0.16) | 0.03 (-0.02 to 0.08) |
| **r7dizz** | Physical functioning: Dizziness as persistent problem | 176 | 2 (1.34 to 2.98)** | 0.11 (0.31) | 0.09 (0.04 to 0.13)*** |
| **r7dresscat** | Binomial: problem getting dressed | 120 | 3.64 (2.39 to 5.54)*** | 0.07 (0.26) | 0.16 (0.11 to 0.2)*** |
| **r7eye** | Sensory problems: Fair or poor eyesight despite use of corrective lenses | | | 2.85 (0.97) | 0.13 (0.09 to 0.18)*** |

*(Continued)*

**Table 3.** (Continued)

| HRS | Definitions variables | N with symptoms for binomial variables | Odds ratios (95% CIs) | Mean (SD) | Correlation coefficients (95% CIs) |
|---|---|---|---|---|---|
| r7fall | fallen down last 2 years | | | 0.75 (2.34) | 0.11 (0.06 to 0.16)*** |
| r7fall_cat1 | More than 1 falls | 513 | 2.2 (1.63 to 2.96)*** | 0.31 (0.46) | 0.13 (0.08 to 0.18)*** |
| r7fall_cat2 | More than 2 falls | 242 | 2.34 (1.65 to 3.31)*** | 0.15 (0.35) | 0.12 (0.07 to 0.17)*** |
| r7frail1_1 | Dizziness as persistent problem, > = 2 falls in previous 2 years, or difficulty lifting 10 pounds | 573 | 3.3 (2.44 to 4.46)*** | 0.35 (0.48) | 0.2 (0.15 to 0.24)*** |
| r7frail1_2 | Weight in wave 2002 minus weight in wave 2004! 10% of weight in wave 2002 or body mass index o18.5 kg/m2 | 94 | 8.49 (5.49 to 13.14)*** | 0.06 (0.23) | 0.27 (0.23 to 0.32)*** |
| r7frail1_3 | Mild to severe cognitive impairment on performance-based measure or according to proxy and interviewer rating | 45 | 5.09 (2.75 to 9.42)*** | 0.03 (0.16) | 0.14 (0.09 to 0.19)*** |
| r7frail1_4 | Fair or poor eyesight despite use of corrective lenses or fair or poor hearing despite use of hearing aides | 645 | 2.24 (1.66 to 3.02)*** | 0.39 (0.49) | 0.13 (0.09 to 0.18)*** |
| r7frail3_2 | Yes to either of two CES-D items: (i) Felt that everything I did was an effort in last week. (ii) Could not get going in last week. | 441 | 9.93 (7.13 to 13.84)*** | 0.27 (0.44) | 0.38 (0.34 to 0.42)*** |
| r7frail3_3 | Frequency of three intensities of activity, lowest quintile (stratified according to sex) | 329 | 17.58 (12.47 to 24.78)*** | 0.2 (0.4) | 0.49 (0.45 to 0.52)*** |
| r7frail3_4 | Time to walk 8 ft, converted to time to walk 15 ft. Cutoff criteria according to sex and height remain the same | 724 | 27.06 (14.95 to 48.96)*** | 0.44 (0.5) | 0.38 (0.34 to 0.42)*** |
| r7frail3_5 | Grip strength: Weakest 20% (stratified according to sex and BMI) | 290 | 12.72 (9.18 to 17.64)*** | 0.18 (0.38) | 0.44 (0.4 to 0.48)*** |
| r7frailim1 | Frailty index: Functional Domains Model | | | 0.83 (0.8) | 0.31 (0.26 to 0.35)*** |
| r7frailim1cat | Frailty status: Functional Domains Model | 340 | 5.91 (4.34 to 8.06)*** | 0.21 (0.41) | 0.3 (0.26 to 0.34)*** |
| r7frailim2 | Frailty index: Burden Model | | | 4.17 (2.07) | 0.27 (0.22 to 0.32)*** |
| r7frailim2cat | Frailty status: Burden Model | | | 0.33 (0.47) | 0.19 (0.14 to 0.24)*** |
| r7frailim3 | Frailty index: Biologic Syndrome Model | | | 1.14 (1.08) | 0.73 (0.7 to 0.75)*** |
| r7grip | Grip strength, largest value | | | 28.31 (13.69) | -0.32 (-0.36 to -0.27)*** |
| r7headac | Headache | 74 | 1.11 (0.56 to 2.2) | 0.05 (0.21) | 0.01 (-0.04 to 0.06) |
| r7hear | Sensory problems: fair or poor hearing despite use of hearing aides | | | 2.88 (1.09) | 0.11 (0.06 to 0.16)*** |
| r7heartcat | Binomial: Cardiac problems | 562 | 1.83 (1.36 to 2.46)*** | 0.34 (0.47) | 0.1 (0.05 to 0.15)*** |
| r7hibp | Had high blood pressure since last interview | 987 | 1.51 (1.11 to 2.07)** | 0.6 (0.49) | 0.06 (0.02 to 0.11)** |
| r7lift | Some difficulty in lift/carry 10lbs | 440 | 6.83 (4.98 to 9.35)*** | 0.27 (0.44) | 0.32 (0.28 to 0.37)*** |
| r7ltactx | Frequency of light physical activity | | | 2.71 (1.06) | 0.25 (0.2 to 0.29)*** |
| r7lungcat | Binomial: Lung problems | 165 | 1.84 (1.21 to 2.79)** | 0.1 (0.3) | 0.07 (0.02 to 0.12)** |
| r7memryscat | Binomial: Memory changes | 20 | 1.79 (0.59 to 5.4) | 0.01 (0.11) | 0.03 (-0.02 to 0.07) |
| r7mobila | Some difficulty in mobility /05 | | | 1.12 (1.38) | 0.37 (0.33 to 0.41)*** |
| r7muscle | Musculoskeletal problems | 57 | 0.53 (0.19 to 1.47) | 0.03 (0.18) | -0.03 (-0.08 to 0.02) |

*(Continued)*

**Table 3.** (Continued)

| HRS | Definitions variables | N with symptoms for binomial variables | Odds ratios (95% CIs | Mean (SD) | Correlation coefficients (95% CIs) |
|---|---|---|---|---|---|
| **r7psychcat** | Binomial: Depression | 165 | 2.38 (1.6 to 3.54)*** | 0.1 (0.3) | 0.11 (0.06 to 0.16)*** |
| **r7psychscat** | Binomial: Changes in general mental functioning | 17 | 1.53 (0.43 to 5.36) | 0.01 (0.1) | 0.02 (-0.03 to 0.06) |
| **r7seizure** | Seizures, generalized | 0 | Not applicable | | Not applicable for uniform values |
| **r7sleeprcat** | Binomial: Sleep changes | 346 | 1.74 (1.25 to 2.41)** | 0.21 (0.41) | 0.08 (0.03 to 0.13)*** |
| **r7strokcat** | Binomial: Cerebrovascular problems | 110 | 2.65 (1.68 to 4.18)*** | 0.07 (0.25) | 0.11 (0.06 to 0.15)*** |
| **r7strokecat** | Binomial: History of stroke | 139 | 2.56 (1.68 to 3.88)*** | 0.08 (0.28) | 0.11 (0.06 to 0.16)*** |
| **r7tired** | Tiredness all the time | 0 | Not applicable | | Not applicable for uniform values |
| **r7toiltcat** | Binomial: Toileting problems | 60 | 6.07 (3.56 to 10.35)*** | 0.04 (0.19) | 0.18 (0.14 to 0.23)*** |
| **r7underw** | Underweight in wave 2004 (%) | 36 | 9.72 (4.95 to 19.09)*** | 0.02 (0.15) | 0.2 (0.15 to 0.24)*** |
| **r7urine** | Urinary incontinence | 360 | 2.14 (1.56 to 2.94)*** | 0.22 (0.41) | 0.12 (0.07 to 0.17)*** |
| **r7walkt15** | Time to walk 15 feet | | | 7.76 (24.91) | 0.42 (0.38 to 0.46)*** |
| **r7wchange** | Weight in wave 2002 minus weight in wave 2004 (%) | | | 0 (0.06) | 0.05 (0 to 0.1)* |
| **r7weight** | Self-reported weight in kilograms | | | 75.75 (16.87) | -0.1 (-0.15 to -0.05)*** |
| **Bias variables** | | | | | |
| **Bias variables induced by the Functional Domains Model** | | | | | |
| **r7frail1_1res** | Bias induced by Dizziness as persistent problem, > = 2 falls in previous 2 years, or difficulty lifting 10 pounds | | | 0 (0.29) | 0.03 (-0.02 to 0.08) |
| **r7frail1_2res** | Bias induced by Weight in wave 2002 minus weight in wave 2004! 10% of weight in wave 2002 or body mass index o18.5 kg/m2 | | | -0.03 (0.22) | 0.21 (0.17 to 0.26)*** |
| **r7frail1_3res** | Bias induced by Mild to severe cognitive impairment on performance-based measure or according to proxy and interviewer rating | | | 0 (0.16) | -0.02 (-0.06 to 0.03) |
| **r7frail1_4res** | Bias induced by Fair or poor eyesight despite use of corrective lenses or fair or poor hearing despite use of hearing aides | | | 0 (0.21) | 0 (-0.05 to 0.05) |
| **Bias variables induced by the Burden Model** | | | | | |
| **r7frail2res** | Bias induced by Summary of proxy memory rating and total cognition summary score | | | 0 (0.01) | -0.21 (-0.27 to -0.16)*** |
| **Bias variables induced by the Biologic Syndrome Model** | | | | | |
| **r7frail3_2res** | Bias induced by Yes to either of two CES-D items: (i) Felt that everything I did was an effort in last week. (ii) Could not get going in last week. | | | 0 (0.17) | 0.08 (0.03 to 0.13)*** |
| **r7frail3_3res** | Bias induced by Frequency of three intensities of activity, lowest quintile (stratified according to sex) | | | 0 (0.31) | 0.34 (0.3 to 0.38)*** |
| **r7frail3_4res** | Bias induced by Time to walk 8 ft, converted to time to walk 15 ft. Cutoff criteria according to sex and height remain the same | | | 0 (0.49) | 0.3 (0.26 to 0.34)*** |
| **r7frail3_5res** | Bias induced by Grip strength: Weakest 20% (stratified according to sex and BMI) | | | 0 (0.34) | 0.26 (0.22 to 0.31)*** |

n = 1,642; frailty n (%) = 540 (32.89%); mean age = 77.05 years; female % = 54.69s%. HRS = Health and Retirement Study.

BMI = body mass index; CES-D = Center for Epidemiological Studies Depression; HRS = Health and Retirement Study.

* = p < 0.05

** = p < 0.01

*** = p < 0.001.

to demonstrate significant correlations with some of the frailty symptoms of their own or those defined by the other two frailty diagnoses. When frailty lacks significant associations with frailty symptoms, this suggests frailty diagnoses are made based on so-called frailty symptoms, some of which frailty may not cause them. This needs serious discussion and examination.

## Why frailty fails to be significantly associated with frailty symptoms?

Frailty diagnoses do not fully support the models of their own by showing insignificant correlations with some of their own frailty symptoms or input variables. One reason may be that frailty researchers did not recognize the importance of causation between frailty and frailty symptoms. The authors of the Burden Model recommended selecting frailty symptoms based on the associations between the candidate symptoms and two factors: age and general health status [18], rather than selecting the symptoms that cause or are caused by frailty. For this model, the so-called frailty symptoms are, in fact, age-related and general health-related variables.

Second, for the Burden Model that requires a large number of frailty symptoms [18], some symptoms may not present in the population at all and are used for frailty diagnosis regardless. For example, in the HRS cohort, since we did not identify any patients with generalized seizures and it was impossible to determine the association between this symptom and the frailty status defined by this model. Including frailty symptoms that do not present in a population can underestimate the prevalence rate of frailty defined by the Burden Model. This is because the diagnostic threshold of this model is proportional to the total number of frailty symptoms used [18]. When a symptom that should not be included in the diagnostic criteria is included, the diagnostic threshold increases with the total number of frailty symptoms. If the symptom, generalized seizures, is excluded from the diagnostic criteria, the diagnostic threshold can decrease and it is likely that more HRS participants can qualify the diagnosis of frailty defined by the Burden Model.

The third reason is that the diagnostic criteria of frailty have been made so complicated that biases have been introduced and interfered the relationships between frailty and frailty symptoms. The biases are produced by censoring the sum of multiple symptoms and dichotomizing continuous variables [2]. These biases are so important that each of the frailty indices defined by the three models is significantly associated with the biases created by their own or the other two models. In fact, the biases generated by the diagnostic criteria of the Biologic Syndrome Model explain this model's frailty index better than its frailty symptoms [2]. When the frailty indices better represent the biases, these indices are less likely to have significant associations with their frailty symptoms.

The last reason is the correlations between frailty symptoms have been neglected. The correlations between symptoms, symptom prevalence, and the design of the diagnostic criteria (whether biases are created and integrated to the diagnosis) are the three major determinants of the prevalence of the diagnosis [24]. Neglecting the importance of symptom correlations can lead to major errors. For example, when five highly correlated variables with the same means (correlation coefficients = 1) are summed to create an index, this index is not very different from five times any one of the five input variables [25]. When two completely opposite variables with mean values of 0 (correlation coefficient = -1) are summed, the derived index contains only 0 [25]. The frailty indices defined by the three frailty models consist of frailty symptoms of various correlations with each other. Some of the frailty symptoms in the three models may be highly correlated with each other. It takes only four frailty symptoms to explain more than 54% of the variances of the three frailty indices that use 9 or more frailty symptoms

[2]. This issue becomes more problematic for frailty diagnoses made based on a large number of frailty symptoms. The frailty status defined by the Burden Model requires at least 30 symptoms for diagnosis and 70 symptoms have often been used [18]. It took only 11 frailty symptoms to explain more than 90% of the variances of the frailty index [2]. The frailty status defined by this model is not significantly associated with two of its frailty symptoms in this study. The role of insignificant symptoms, seizure and tiredness, in the frailty status defined by the Burden Model haven't been discussed in previous studies [18].

## Causation?

In addition to the strengths of associations, the evidence to support the causal relationship between frailty and frailty symptoms seems limited. The pathological changes that are considered related to frailty include sarcopenia, heart disease, and lung disease, depending on the frailty models [2]. In this study, grip strength, cardiac problems, and lung problems were significantly correlated with the three frailty indices, respectively. However, the biological and pathological evidence that support the causal relationship between frailty and the frailty symptoms of the other organ systems seems insufficient [26].

## Assumptions

The different patterns of the insignificance between frailty symptoms and the 3 frailty statuses indicated conflicting views on frailty. The 3 frailty models have major discrepancies in the underlying assumptions, including theoretical frameworks, age thresholds, the selection of frailty symptoms, and the design of diagnostic criteria [2]. Subsequently, the differences in these assumptions between frailty models can be shown with the symptoms that best explained frailty statuses [2]. We found the frailty symptoms or input variables that had the largest correlation coefficients with the three frailty statuses were different. The three symptoms that have the largest significant correlation coefficients with the frailty indices defined by the Functional Domain Model, the Burden Model, and the Biologic Syndrome Model are some difficulties in lifting 10 pounds, some difficulty in mobility, and slowness measured by time to walk 8 feet, respectively. The paths from non-frailty to frailty vary depending on the models used.

## Logic challenges

These results highlight logic challenges. Frail patients are not more likely to have certain frailty symptoms, but these symptoms are necessary to make these diagnoses. It is unclear whether the symptoms that frailty is insignificantly correlated with can be called "frailty" symptom or used for frailty diagnosis. When excluding these symptoms from being used for the diagnosis of frailty, the prevalence of frailty will likely decrease for the Functional Domain Model and the Biologic Syndrome Model and may increase or decrease for the Burden Model. Whether the updated indices will become insignificantly associated with other frailty symptoms is unclear. If the frailty symptoms that are not significantly associated with frailty should be excluded, many of the published frailty prevalence rates are likely to be overestimated or biased for the reason describe above. We have not identified any studies explicitly examine the significance of the associations between the frailty statuses and frailty symptoms they defined in their own models. We will continue exploring the causal relationship between frailty and frailty symptoms using other data sets in the future.

## Limitations

This study has strengths in using a publicly accessible database that has been investigated in previous studies [1, 2]. The demographic characteristics reported in this study matched those

reported [1]. However, there are several limitations to this study. There are other statistical and epidemiological measures of association that can be tested to demonstrate the strengths of associations, including Chi-squared statistics and risk ratios [27, 28]. Odds ratios are adequate for cross-sectional studies to approximate risk ratios [17]. However, odds ratios can over- or under-estimate effect sizes if the underlying risk ratios are greater or less than 1, respectively [29]. Other measures of association will be explored in the future. Moreover, there are other factors influencing the correlations between frailty statuses and frailty symptoms, such as demographic characteristics. These factors can be adjusted using techniques, such as multiple regression [30, 31]. Lastly, this study used cross-sectional data and longitudinal follow-up of the strengths of the associations between frailty and frailty symptoms might help to answer important questions, such as whether the insignificant associations are transient, whether frailty predicts major outcomes better if insignificant frailty symptoms are discarded, and whether the biases induced by the frailty diagnostic criteria predict outcomes better than frailty symptoms. This will need to be explored in future research.

## Conclusion

The frailty diagnoses defined by three models were assessed for their correlations with frailty symptoms of their own, those defined by the other two models, and bias variables using odds ratios and correlation coefficients. Frailty diagnoses lack significant correlations with some of their own frailty symptoms and some of the frailty symptoms defined by the other two models. This suggests that frail patients are not more likely to have certain frailty symptoms using any of the three frailty models. This finding raises questions like whether frailty symptoms lacking significant correlations with frailty statuses could be included to diagnose frailty and whether frailty exists and causes frailty symptoms. Further research to assess the causal relationships between frailty and frailty symptoms is needed and planned.

## Author Contributions

**Conceptualization:** Yi-Sheng Chao.

**Data curation:** Yi-Sheng Chao, Chao-Jung Wu.

**Formal analysis:** Yi-Sheng Chao.

**Investigation:** Yi-Sheng Chao.

**Methodology:** Yi-Sheng Chao.

**Project administration:** Yi-Sheng Chao.

**Resources:** Yi-Sheng Chao.

**Software:** Yi-Sheng Chao.

**Supervision:** Yi-Sheng Chao.

**Validation:** Yi-Sheng Chao.

**Visualization:** Yi-Sheng Chao.

**Writing – original draft:** Yi-Sheng Chao.

**Writing – review & editing:** Yi-Sheng Chao, Chao-Jung Wu, June Y. T. Po, Shih-Yu Huang, Hsing-Chien Wu, Hui-Ting Hsu, Yen-Po Cheng, Yi-Chun Lai, Wei-Chih Chen.

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
