## [Decision Letter · Decision Letter 0]

10 May 2022

PONE-D-21-33729Frailty does not cause all frail symptoms: United States Health and Retirement StudyPLOS ONE

Dear Dr. Chao,

Thank you for submitting your manuscript to PLOS ONE. After careful consideration, we feel that it has merit but does not fully meet PLOS ONE’s publication criteria as it currently stands. Therefore, we invite you to submit a revised version of the manuscript that addresses the points raised during the review process.

The two reviewers made substantial comment on the paper, the first being more food for thought than substantial request for revision, but worthy of consideration and comment

We look forward to receiving your revised manuscript.

Kind regards,

Adrian Stuart Wagg, MD

Academic Editor

PLOS ONE

Journal Requirements:

YSC is employed by the Canadian Agency for Drugs and Technologies in Health. YSC conducted this study as an independent researcher out of academic curiosity without any material support. No external funding was received for this study. This study is not associated with any patents, products in development or marketed products.

Reviewers' comments:

Reviewer's Responses to Questions

**Comments to the Author**

1. Is the manuscript technically sound, and do the data support the conclusions?

Reviewer #1: Yes

Reviewer #2: Yes

2. Has the statistical analysis been performed appropriately and rigorously? 

Reviewer #1: I Don't Know

Reviewer #2: Yes

3. Have the authors made all data underlying the findings in their manuscript fully available?

Reviewer #1: Yes

Reviewer #2: Yes

4. Is the manuscript presented in an intelligible fashion and written in standard English?

Reviewer #1: Yes

Reviewer #2: Yes

5. Review Comments to the Author

Reviewer #1: It is always to see research done to appreciate and explore the concept of fitness of frailty and to explore which aspects of this phenomenon can potentially be intervened on.

However; I am not sure if I agree with the premise of this paper, which seems to assume that each variable that makes up a model of measurement of physiological reserves or “frailty”, ought to correlate with the entire model or measuring tool that has been described. I am also not entirely surprised to see that there are some variables, when assessed individually, that do not correlate with the models or measures of frailty.

Comments:

1. “Frailty is a geriatric syndrome….”. This is not true. Frailty is a concept/phenomenon that has been shown to exist across the numeric age spectrum. Although it is increasingly prevalent with increasing numeric age, it is not an exclusive phenomenon in those who are of a particular older numeric age.

2. Frailty is a term used to describe the amount or degree of physiological reserves an individual has. Such that, the more amount of physiological reserves an individual has, the more resilient he or she is in the way of any “stressors” (e.g. disease, interventional treatments) that they face. It takes less degree of stressor to destabilize an individual’s homeostasis, if they have less reserves or if they are frail. Fit patients are less likely to decompensate medically and functionally in the way of stressors, unlike frail individual, because they more physiological reserves.

Fitness and frailty are used to describe the entire set of reserves in an individual. It is akin to trying to determine the degree of fibrotic tissue or liver reserves or cirrhosis in an individual with liver disease. However, “frailty” is more complex, because it is an attempt describe and measure reserves in an entire individual

3.regarding the line: “ the causal relationships between frailty and frailty symptoms can be confirmed based on previously published criteria.” I am not convinced that these criteria or any other attempts to measure or describe the concept of physiological reserves or frailty, has demonstrated any casual relationship between frailty or any “frailty symptoms as is suggested by this paper.

4. Instead, the ways in which various measures or tools that are used to assess frailty, use a multitude of variables that include:

a. Contributing factors (e.g. various disease processes, comorbidities)

b. Phenotypic markers (e.g. sarcopenia, and measures of sarcopenia – e.g. grip strength)

c. Sequalae of being fit or frail (e.g. degree or function or dysfunction, number or falls, low mood)

To what extent an individual’s reserves are diminished cannot accurately be determined by any single variable captured by a tool. Simply having the presence of heart failure, does not automatically reflect that someone is frail. However, in someone with long standing severe heart failure, it is conceivable that they may become deconditioned after some time (partially reflected by sarcopenia, and it’s measures) , and that after some time, that degree of sarcopenia or deconditioning may lead to loss of strength and decline in functioning. It is conceivable that after some time, that they may have low mood or depression as a result of their inability to function to their previous extent.

At the same time, it is also conceivable that a disorder such as a depression may not only be a resulting factor of being frail, but it may also be a contributing factor. For example, someone with a primary psychiatric diagnosis of major depressive disorder, with otherwise manageable heart failure, may not adhere to their medications while depressed. They may also not be physically active. This may worsen their heart failure. This may in turn lead to them to becoming frail, if their issues are not managed.

5. Single variables may also be mild, early, or transient (potentially cured) in nature, such that their

existence at a particular cross-section (when the frailty measure is use) may not be enough to cause a degree of diminished physiological reserves to be frail. This is conceivable for the variables assessed in this study, For example, malignant disease or history of, tiredness all of the time, memory change and headache, which did not correlate with frailty status. Some variables, when they become severe, may draw out other variables that are part of a model or measurement tool for frailty/reserves.

In that regard, a single variable may not capture the extent of diminished reserves, as perhaps a tool or measure that encompasses multiple variables does or a tool that captures the enduring effects the things that can contribute to frailty do (e.g. permanent loss of function). As such, the interaction of the variables also plays an important role.

6. Other specific aspects of the paper

a. It would be easier to label the measure or models as they are commonly described e.g. Rockwood’s “frailty index” or “Fried model” or “Phenotypic model”. Otherwise, it does get confusing to keep track of which is what with newer ways of describing them.

b. The paper by Strawbridge et al. that was published in 1998 lays out variables or items that are more commonly reflected in those who are frail versus non frail. I am not sure if it was intended to be a “model” or a measure of reserves, in the way that the other two were. Also, has this approach been validated in various settings as has the other two models?

c. The use of the term “frailty symptoms”, in my opinion, over simplifies what the variables in the various measures of physiological reserves/fitness-frailty are attempting to capture. (see point 4 above).

d. “it has been suggested that cognitive impairment plays an important role for frailty diagnosis and mortality among frail patients”. Cognitive impairment, if it does not go beyond erosion of domain deficits, that is if there is no resulting functional deficits (which would then be dementia) may reflect some degree of diminishment physiological reserves, but it may not reach the “threshold” for reflecting frailty in the way moderate to severe might. At the same time, there are individuals who have enough of their physiological reserves eroded (due to other processes or variables) that they have become “frail”, but who still have their cognitive abilities intact.

Overall:

Areas of improvements or cons of the article: There is an over simplification of the concept of frailty reflected by terms such as: “diagnose frailty”, “symptoms of frailty” and “causality” .

I am also not entirely convinced on the premise of the article which seems to be that the individual variables that make up the various tools ought to be reflective the extent of physiological reserves or frailty and be as reflective of the ability of the various measuring models or tools (that capture a variety of variables).

Items/variables used in the various models or measuring tools of assessing physiological reserves/fitness-frailty do not necessarily reflect "symptoms of frailty"(see point 4). In addition, variables that reflect degree of physiological reserves (either as contributing factors, phenotypic manifestations or resulting factors) can either be mild or severe or transient, and can draw in other variables given the nature of the variable.

As such, a more nuanced approach is required to assess which variables that reflect frailty, can be "targeted" to help management strategies. For example: in this article, malignancy or having a history of malignancy did not correlate with being frail. Once could conceive that if an early malignant process were to progress towards metastasis, that perhaps at some point, a patient would be more “frail” towards the end of the process than at the start. It would be worthwhile trying to stop the malignancy nonetheless, as it would be important to try to stop or modify other contributing factors to frailty, before someone has already become frail.

Positives: Exploring models or measures of “reserves” to determine their strengths and weakness.

There is also, I believe, an unintended, but important purpose that can be derived from this article, even though it has not been stated: in an effort to choose a measuring tool or model for “physiological reserves” or frailty, there may be value in having a tool that is practical to use in particular settings. That practically may be achieved with a tool that perhaps has the least amount of variables, while still being effective.

Reviewer #2: Please refer to the attached document for my comments. Overall, I think this an interesting and valuable study, which has highlighted an important issue. All my comments are all focused on spelling and grammar only.

6. PLOS authors have the option to publish the peer review history of their article (what does this mean?). If published, this will include your full peer review and any attached files.

Reviewer #1: No

Reviewer #2: No

---

## [Author Response · Author response to Decision Letter 0]

15 Jun 2022

Reply to the reviewers

Reviewer #1: It is always to see research done to appreciate and explore the concept of fitness of frailty and to explore which aspects of this phenomenon can potentially be intervened on. 

However; I am not sure if I agree with the premise of this paper, which seems to assume that each variable that makes up a model of measurement of physiological reserves or “frailty”, ought to correlate with the entire model or measuring tool that has been described. I am also not entirely surprised to see that there are some variables, when assessed individually, that do not correlate with the models or measures of frailty. 

Authors’ reply: thank you for the comments and all your time and efforts to help improve this manuscript. We appreciated your inputs and were happy that you were fine with the fact that not all input variables were significantly associated with frailty. We were not taking a position for whether frailty should be significantly associated with frailty symptoms or not. In the Introduction, we added that we were uncertain whether frailty should cause frailty symptoms. Different people will have very different opinions about the causal relationship between them. We presented the results only to demonstrate the lack of association between frailty and some frailty symptoms. We greatly appreciated the opportunity to have this in-depth conversation with an experienced frailty researcher like you.

Comments: 1. “Frailty is a geriatric syndrome....”. This is not true. Frailty is a concept/phenomenon that has been shown to exist across the numeric age spectrum. Although it is increasingly prevalent with increasing numeric age, it is not an exclusive phenomenon in those who are of a particular older numeric age. 

Authors’ reply: thank you for the correction. We agreed and removed “geriatric”. We did find that more than 10% of the populations younger than the age thresholds of 3 commonly used frailty models were considered frail in a previous study (page 7, Table 1 in https://journals.plos.org/plosone/article?id=10.1371/journal.pone.0197859). Frailty has been found prevalent in many age groups.

2. Frailty is a term used to describe the amount or degree of physiological reserves an individual has. Such that, the more amount of physiological reserves an individual has, the more resilient he or she is in the way of any “stressors” (e.g. disease, interventional treatments) that they face. It takes less degree of stressor to destabilize an individual’s homeostasis, if they have less reserves or if they are frail. Fit patients are less likely to decompensate medically and functionally in the way of stressors, unlike frail individual, because they more physiological reserves. 

Fitness and frailty are used to describe the entire set of reserves in an individual. It is akin to trying to determine the degree of fibrotic tissue or liver reserves or cirrhosis in an individual with liver disease. However, “frailty” is more complex, because it is an attempt describe and measure reserves in an entire individual 

Authors’ reply: thank you for the introduction to the frailty theory. We agreed that physiological reserves might have been the target that various frailty models aimed to measure. However, many might fail to measure the physiological reserves they hoped to quantify. We interpreted 3 commonly used frailty indices using frailty symptoms and found frailty indices could be understood as frailty symptoms plus different weights on them, rather than physiological reserves. Our findings suggested that it took few frailty symptoms to explain frailty indices (page 8, Table 1 in https://journals.plos.org/plosone/article?id=10.1371/journal.pone.0197859). For example, the frailty index defined by the Burden model by Rockwood et al. (2007) that required 70 variables to measure physiological reserve could be mostly explained by 4 frailty symptoms: 1) History of stroke; 2) Depression (clinical impression); 3) Impaired mobility; 4) Urinary incontinence (R-squared = 0.667). This is to say that determining the degree of frailty an individual (frailty index range: 0 to 1 based on 70 variables) has is not very different from asking whether this individual has a history of stroke, depression, impaired mobility, and urinary incontinence. This is just one of the 3 frailty models. The other frailty models had their own frailty symptoms that best explained them. We were uncertain about whether these 4 variables represented the physiological reserve that a frailty index aimed to measure. We agreed that frailty is complex, but the measurement of frailty is adding more complexity for the understanding of frailty.

3.regarding the line: “ the causal relationships between frailty and frailty symptoms can be confirmed based on previously published criteria.” I am not convinced that these criteria or any other attempts to measure or describe the concept of physiological reserves or frailty, has demonstrated any casual relationship between frailty or any “frailty symptoms as is suggested by this paper. 

Authors’ reply: thank you. We agreed that causation could be conceived very differently. We revised the texts to show there are different ways to establish causal relationships. We thought the diversity in the understanding in causation very important and even did a survey to understand how people thought about the causal relationship between diagnosis and symptoms (Table 3 in https://www.frontiersin.org/articles/10.3389/fpsyt.2022.860487/full). Some (7.14%) agreed with you that the causation of symptoms by diagnoses could be confirmed by including the symptoms in the diagnostic criteria. However, many more valued other approached more for causal relationships. 28.57% and 57.14% considered strengths of associations and pathological evidence, respectively, important to confirm causation between diagnoses and symptoms.

4. Instead, the ways in which various measures or tools that are used to assess frailty, use a multitude of variables that include: 

a. Contributing factors (e.g. various disease processes, comorbidities) b. Phenotypic markers (e.g. sarcopenia, and measures of sarcopenia – e.g. grip strength) c. Sequalae of being fit or frail (e.g. degree or function or dysfunction, number or falls, low mood) 

To what extent an individual’s reserves are diminished cannot accurately be determined by any single variable captured by a tool. Simply having the presence of heart failure, does not automatically reflect that someone is frail. However, in someone with long standing severe heart failure, it is conceivable that they may become deconditioned after some time (partially reflected by sarcopenia, and it’s measures) , and that after some time, that degree of sarcopenia or deconditioning may lead to loss of strength and decline in functioning. It is conceivable that after some time, that they may have low mood or depression as a result of their inability to function to their previous extent. 

At the same time, it is also conceivable that a disorder such as a depression may not only be a resulting factor of being frail, but it may also be a contributing factor. For example, someone with a primary psychiatric diagnosis of major depressive disorder, with otherwise manageable heart failure, may not adhere to their medications while depressed. They may also not be physically active. This may worsen their heart failure. This may in turn lead to them to becoming frail, if their issues are not managed. 

Authors’ reply: thank you very much for the introduction to frailty symptom classifications. We agreed that frailty symptoms might represent different relationships with frailty. However, the diagnostic criteria of 3 commonly used frailty indices (the Functional Domain model proposed by Strawbridge et al. (1998), the Burden model by Rockwood et al. (2007), and the Biological Syndrome model by Fried et al. (2004)) were not designed properly and distorted the relationships between frailty and frailty symptoms (see the visualization in Fig 2 in https://journals.plos.org/plosone/article?id=10.1371/journal.pone.0197859). In the same article, we noticed that some frailty symptoms were weighted much more than the magnitude we expected from reading the diagnostic criteria. Single frailty symptoms could be in fact much more important than the others, ie explaining most of the frailty index variances. For example, the Burden model adopted too many variables related to cardiovascular disease and the single frailty symptom that best interpreted the frailty index was a history of stroke (Table 1 in page 8, https://journals.plos.org/plosone/article?id=10.1371/journal.pone.0197859). This overemphasis of single symptoms due to the distortion created by the diagnostic criteria contradicted the heart failure example. We found having a history of stroke could significantly predict frailty status defined by the Burden Model.

5. Single variables may also be mild, early, or transient (potentially cured) in nature, such that their existence at a particular cross-section (when the frailty measure is use) may not be enough to cause a degree of diminished physiological reserves to be frail. This is conceivable for the variables assessed in this study, For example, 

malignant disease or history of, tiredness all of the time, memory change and headache, which did not correlate with frailty status. Some variables, when they become severe, may draw out other variables that are part of a model or measurement tool for frailty/reserves. 

In that regard, a single variable may not capture the extent of diminished reserves, as perhaps a tool or measure that encompasses multiple variables does or a tool that captures the enduring effects the things that can contribute to frailty do (e.g. permanent loss of function). As such, the interaction of the variables also plays an important role. 

Authors’ reply: thank you for the hypotheses for the lack of significant associations between frailty symptoms and frailty. They are great research topics for longitudinal data. However, our previous analysis of the same data revealed reasons why some frailty symptoms were not significantly associated with frailty indices. The diagnostic criteria of 3 frailty indices were overly complicated and encouraged inadequate data processing. There were 2 main mechanisms that aggregated information from input symptoms and introduced biases to the diagnoses: censoring of multiple variables and dichotomization of continuous variables (see the visualization in Fig 2 in https://journals.plos.org/plosone/article?id=10.1371/journal.pone.0197859). In this article, we reproduced the 3 frailty indices created by other researchers and derived the bias variables that were created by information censoring of multiple variables or dichotomization of continuous variables. There were 4, 1, and 5 sources of biases identified for the 3 frailty models (Table 1 in the same article). These bias variables were used to interpret frailty indices, compared to frailty symptoms. The results were incredible to many. The frailty index defined by Fried’s Biologic Syndrome Model was better explained by biases (information undesired, produced by inadequate data processing) than by its input frailty symptoms (Table 1 in the above-mentioned article). This clearly showed that biases explained the information in 3 commonly used frailty indices well, sometimes better than frailty symptoms. The data processing embedded in the complicated diagnostic criteria for the diagnosis of frailty not only created bias variables, but also weakened the connection between frailty symptoms and frailty indices. This fact could be supported by the R-squared of frailty symptoms and bias variables to interpret frailty indices, as well as the area under ROCs of frailty symptoms and bias variables to predict binominal frailty statuses (Table 1 in the article). For the Rockwood’s Burden Model, it aggregated too many variables to create a frailty index. Imaging summing 69 cardiovascular disease related symptoms (diastolic hypertension, systolic hypertension, history of hypertension, history of abnormal lipid profile,…) and 1 unrelated symptom (eg tiredness) into an index, this index could be very well explained by 1 or 2 CV symptoms because these CV symptoms were highly correlated and having 1 or 2 of the CV symptoms could explain much of the variances of the new index. However, the unrelated symptoms, tiredness, would be very likely to have insignificant association with the index because this index, in fact, represented CV symptoms. In Table 1 in the above-mentioned article, using the Health and Retirement Study, the frailty index defined by the Burden Model could be explained by 11 frailty symptoms, R-squared > 0.9. Using many correlated symptoms for this model could lead to the insignificant association between frailty index and some frailty symptoms.

Some transient effects or other mechanisms you hypothesized might partly explain the lack of significant associations between some frailty symptoms and frailty indices in longitudinal analysis. However, the evidence to show the inadequate data processing embedded in the frailty diagnostic criteria was strong even in cross-sectional data and existing in 3 commonly used frailty indices.

6. Other specific aspects of the paper a. It would be easier to label the measure or models as they are commonly described e.g. Rockwood’s “frailty index” or “Fried model” or “Phenotypic model”. Otherwise, it does get confusing to keep track of which is what with newer ways of describing them. 

Authors’ reply: thank you for the comment. We agreed that frailty models were named or called very differently. In a previously published article (https://journals.plos.org/plosone/article?id=10.1371/journal.pone.0197859), we followed the terminology that Cigolle et al. (2009) used to describe 3 frailty models and analyze the Health and Retirement Study. Because the results were not supportive to the 3 frailty models and there were a lot of critiques to these models, we thought it better to label these models without directly naming the researchers.

b. The paper by Strawbridge et al. that was published in 1998 lays out variables or items that are more commonly reflected in those who are frail versus non frail. I am not sure if it was intended to be a “model” or a measure of reserves, in the way that the other two were. Also, has this approach been validated in various settings as has the other two models? 

Authors’ reply: thank you for this point. We searched for some time and found only 1 article (Cigolle et al. 2009: https://agsjournals.onlinelibrary.wiley.com/doi/10.1111/j.1532-5415.2009.02225.x) that used patient-level data and compared more than 2 frailty models. We followed Cigolle’s terminology and called Strawbridge et al.’s frailty definition as “the Functional Domains Model”. We thought Cigolle et al. were satisfied with the validity of this model and its agreement with Rockwood’s and Fried’s models.

c. The use of the term “frailty symptoms”, in my opinion, over simplifies what the variables in the various measures of physiological reserves/fitness-frailty are attempting to capture. (see point 4 above). 

Authors’ reply: thank you for the clarification. We agreed that these symptoms might represent different components of frailty or physiological reserves, as you pointed out here and above. We began to call these input variables as frailty symptoms in our syndrome mining article (https://www.nature.com/articles/s41598-020-60869-8), because we did not find evidence to show these symptoms could be categorized in ways you described above. We tried all possible combinations of any 4 frailty symptoms to find the patterns of significant frailty indices. These symptoms were ordered by the effect sizes of predicting mortality. We found the key to produce a statistically significant frailty index with a large effect size for mortality prediction is to add a frailty symptom with a large effect size, “cognitive impairment”, and 3 insignificant frailty symptoms with near-zero regression coefficients. Creating a significant frailty index is simple with a high success rate, > 99%, if you have many age-related symptoms. Creating a significant frailty index does not seem to be related to the categories that researchers use to group frailty symptoms.

d. “it has been suggested that cognitive impairment plays an important role for frailty diagnosis and mortality among frail patients”. Cognitive impairment, if it does not go beyond erosion of domain deficits, that is if there is no resulting functional deficits (which would then be dementia) may reflect some degree of diminishment physiological reserves, but it may not reach the “threshold” for reflecting frailty in the way moderate to severe might. At the same time, there are individuals who have enough of their physiological reserves eroded (due to other processes or variables) that they have become “frail”, but who still have their cognitive abilities intact. 

Authors’ reply: thank you for the comment. We agreed that cognitive impairment may interact with other frailty symptoms. However, we found cognitive impairment was an independent risk factor for mortality because it lacked significant interactions with other frailty symptoms defined by the Functional Domains Model (https://www.nature.com/articles/s41598-020-58782-1). This finding was based on a survival analyses we conducted based on frailty symptoms and the interactions between these symptoms. The interactions between frailty symptoms did not significantly predict mortality and we concluded the frailty symptoms defined by Functional Domains Model independently predicted mortality (Figure 7 in https://www.nature.com/articles/s41598-020-58782-1).

Overall: Areas of improvements or cons of the article: There is an over simplification of the concept of frailty reflected by terms such as: “diagnose frailty”, “symptoms of frailty” and “causality” . 

Authors’ reply: thank you for the introduction to frailty theory. We used terms that we used in previous publications that aimed to interpret frailty. For example, we diagnosed frailty in an article that interpreted frailty with frailty symptoms (https://journals.plos.org/plosone/article?id=10.1371/journal.pone.0197859). We defined frailty symptoms in a syndrome mining article (https://www.nature.com/articles/s41598-020-60869-8). We discussed causality between symptoms and diagnoses in another article (https://www.frontiersin.org/articles/10.3389/fpsyt.2022.860487/full). We added these references to the texts. Hope this works.

I am also not entirely convinced on the premise of the article which seems to be that the individual variables that make up the various tools ought to be reflective the extent of physiological reserves or frailty and be as reflective of the ability of the various measuring models or tools (that capture a variety of variables). 

Authors’ reply: thank you for the comment. This is philosophical. This is a hypothesis-generating manuscript. In the Abstract and Conclusion, we left questions like whether the insignificant associations should be considered for frailty symptom selection. We revised the texts to make this point clear and ensure that this manuscript did not claim that symptom-based diagnoses “should cause” symptoms. We want this manuscript to describe the fact that frailty was not significantly associated with some frailty symptoms (consistent with the terminology in a previous publication: https://www.nature.com/articles/s41598-020-60869-8). We already explored the reasons why frailty did not have significant associations with frailty symptoms in another publication (https://journals.plos.org/plosone/article?id=10.1371/journal.pone.0197859). The reasons were described in detail and added to the Discussion. When frailty symptoms were aggregated into frailty indices, information from the symptoms was top-censoring or dichotomized and biases were introduced to frailty indices.

Items/variables used in the various models or measuring tools of assessing physiological reserves/fitness-frailty do not necessarily reflect "symptoms of frailty"(see point 4). In addition, variables that reflect degree of physiological reserves (either as contributing factors, phenotypic manifestations or resulting factors) can either be mild or severe or transient, and can draw in other variables given the nature of the variable. 

As such, a more nuanced approach is required to assess which variables that reflect frailty, can be "targeted" to help management strategies. For example: in this article, malignancy or having a history of malignancy did not correlate with being frail. Once could conceive that if an early malignant process were to progress towards metastasis, that perhaps at some point, a patient would be more “frail” towards the end of the process than at the start. It would be worthwhile trying to stop the malignancy nonetheless, as it would be important to try to stop or modify other contributing factors to frailty, before someone has already become frail. 

Authors’ reply: thank you for rephrasing the comment in Point 5. We hope the reply to Point 5 also addresses this comment well. We also added longitudinal analysis as one of the future projects in the Limitations.

Positives: Exploring models or measures of “reserves” to determine their strengths and weakness. 

There is also, I believe, an unintended, but important purpose that can be derived from this article, even though it has not been stated: in an effort to choose a measuring tool or model for “physiological reserves” or frailty, there may be value in having a tool that is practical to use in particular settings. That practically may be achieved with a tool that perhaps has the least amount of variables, while still being effective. 

Authors’ reply: thank you for the comment. We think frailty diagnoses could be improved in various ways, for example using a scale we created based on patient perspectives without using composite diagnostic criteria (https://bmcresnotes.biomedcentral.com/articles/10.1186/s13104-019-4206-3). We agreed that frailty measures could be simplified and already proved this point in a previous publication (Table 1 in https://journals.plos.org/plosone/article?id=10.1371/journal.pone.0197859). More than 54.1% of the variances of the 3 frailty indices could be explained by 4 frailty symptoms. It took 11 frailty symptoms to explain more than 90% of the variances of the frailty index defined by Rockwood’s Burden Model (that usually took 70 variables). The 3 frailty indices were too complicated and their diagnostic criteria introduced biases that distorted the relationships between frailty indices and frailty symptoms. We appreciated all your time and efforts to help improve this manuscript. The conversation with an experienced frailty expert is a great learning experience for us. Thank you.

Reviewer #2: Please refer to the attached document for my comments. Overall, I think this an interesting and valuable study, which has highlighted an important issue. All my comments are all focused on spelling and grammar only. 

Authors’ reply: thank you for the comment and the detailed review. We appreciated your input. Corrections were made based on your comments.

---

## [Decision Letter · Decision Letter 1]

18 Jul 2022

Frailty does not cause all frail symptoms: United States Health and Retirement Study

PONE-D-21-33729R1

Dear Dr. Chen,

We’re pleased to inform you that your manuscript has been judged scientifically suitable for publication and will be formally accepted for publication once it meets all outstanding technical requirements.

Kind regards,

George Vousden

Staff Editor

PLOS ONE

Additional Editor Comments (optional):

Reviewers' comments:

Reviewer's Responses to Questions

**Comments to the Author**

1. If the authors have adequately addressed your comments raised in a previous round of review and you feel that this manuscript is now acceptable for publication, you may indicate that here to bypass the “Comments to the Author” section, enter your conflict of interest statement in the “Confidential to Editor” section, and submit your "Accept" recommendation.

Reviewer #1: All comments have been addressed

Reviewer #2: All comments have been addressed

2. Is the manuscript technically sound, and do the data support the conclusions?

Reviewer #1: Yes

Reviewer #2: Yes

3. Has the statistical analysis been performed appropriately and rigorously? 

Reviewer #1: I Don't Know

Reviewer #2: Yes

4. Have the authors made all data underlying the findings in their manuscript fully available?

Reviewer #1: Yes

Reviewer #2: Yes

5. Is the manuscript presented in an intelligible fashion and written in standard English?

Reviewer #1: Yes

Reviewer #2: Yes

6. Review Comments to the Author

Reviewer #1: (No Response)

Reviewer #2: I think this is an interesting study which has highlighted a controversial issue that should be looked at.

7. PLOS authors have the option to publish the peer review history of their article (what does this mean?). If published, this will include your full peer review and any attached files.

Reviewer #1: **Yes: **Naheed A. Rajabali MD MSc FRCPC

Reviewer #2: No

---

## [Editor Report · Acceptance letter]

24 Oct 2022

PONE-D-21-33729R1 

Frailty does not cause all frail symptoms: United States Health and Retirement Study 

Dear Dr. Chen:

I'm pleased to inform you that your manuscript has been deemed suitable for publication in PLOS ONE. Congratulations! Your manuscript is now with our production department. 

Kind regards, 

on behalf of

Dr. George Vousden 

Staff Editor

PLOS ONE